# The Role of Sediment Records in Environmental Forensic Studies: Two Examples from Italy of Research Approaches Developed to Address Responsibilities and Management Options

Luca Giorgio Bellucci * and Silvia Giuliani

Institute of Marine Sciences, National Research Council of Italy, 40129 Bologna, Italy; silvia.giuliani@bo.ismar.cnr.it
* Correspondence: luca.bellucci@bo.ismar.cnr.it; Tel.: +39-0516398851

**Abstract:** The coupling of scientific evidence from sediment cores with historical information represents an effective way to reconstruct and quantify recent anthropogenic impacts in transitional and marine-coastal areas. These are both key points pertaining to studies that aim at establishing the responsibility for environmental pollution. Good practices for the selection of sampling sites and specific survey techniques are fundamental to understand pollution histories and dynamics, together with reliable dating methods and analytical procedures. In addition, a certain degree of flexibility and willingness to explore different research pathways is necessary, particularly when unexpected questions arise from scientific data or from requests posed by authorities in charge of preliminary investigations or court debates. In this paper, two different study cases are reviewed, and the approaches developed to tackle with specific issues are presented. Its main purpose is both to explain study paths undertaken to answer challenging scientific-legal questions and to provide examples for developing countries that present similar risks of uncontrolled industrialization. Results were used in preliminary investigations or court debates for the attribution of responsibility for environmental pollution to past or present industrial managements. In addition, they were fundamental for other studies aiming at implementing models that simulate the fate and distribution of contaminants and human exposure. In the Augusta Harbor, an integrated approach merged archive information, bathymetry, and high-resolution seismic profiles with the results of an independent tracer (hexachlorobenzene, HCB). This approach helped attribute the presence of high surficial Hg concentrations to resuspension and redistribution of deep sediments caused by dredging and maritime traffic and not to active outfalls. In the Venice Lagoon, an extensive literature search supported analytical results for the correct identification of industrial processes responsible for the contamination by polychlorinated dibenzo-p-dioxins and furans (PCDD/Fs) in the canals of the Porto Marghera Industrial Area. In addition, sedimentary profiles of PCDD/Fs in local salt marshes ("barene") recorded well the events relative to the industrial development and management of the area reported by historical documents, confirming their potential for this kind of investigation.

**Keywords:** sediment records; management; environmental forensic; industrial pollution; Venice Lagoon; Augusta Bay



## 1. Introduction

Industrial activities and urban settlements (mostly located close to seas and freshwater bodies) have the potential to discharge contaminants in great quantities through their wastes, especially when major environmental laws or control procedures are missing or insufficient [1–5]. Scientists can then be interested in assessing both contamination levels and input chronologies of such endangered areas. This interest might result not only from curiosity-driven motivations, but as a specific request to provide expert contribution to

policy and decision makers who are in charge of health security measures and/or need to assess the responsibility of past and present industrial managements [6–8].

Accreting aquatic sediments can be extremely important sources of information, mostly because they have the potential to behave as natural archives through the retention of significant information about the surrounding environment at the time of their deposition and accumulation [6,9–14]. Specifically, their role as recorders of past chemical signatures makes them ideal matrices for the reconstruction of relatively recent anthropogenic impacts [15–21]. Indeed, the comparison with historical information allows the attribution of observed changes along the core profile to specific activities/management choices. This is possible when geochronological studies supply reliable analytical results [22,23]. It is evident that this kind of information is highly relevant in environmental forensic studies, in particular during court debates that aim at recognizing when environmental pollution started, how it changed, and who has to be held responsible for it. However, sampling strategies should be carefully designed and carried out, in order to avoid the use of inappropriate techniques that can partially or totally obscure the potential information preserved in sediments. For example, sampling devices need to be carefully chosen, since the collection of undisturbed sedimentary profiles is a mandatory request for studies that deal with historic reconstructions of events and changes in the dynamic of the environment. When used in forensic studies that deal with recent events, gravity coring systems are to be favoured with respect to vibro-corers: the first enable the preservation of the sediment water interface, thus ensuring the collection of the recent-most depositions and the least disturbance to subsurface sediment layers. However, gravity coring systems are generally not able to collect sandy sediments that then act as an impenetrable barrier for the sampling of deeper layers [6]. Identifying suitable sampling sites is then an important critical step and is generally obtained through bathymetry and high-resolution seismic surveys prior to sampling campaigns [6,24–26]. These surveys are able to reveal areas where sediment accumulation is more regular or where bottom morphology is affected by slumping, mass accumulation, presence of gas, etc. [6,24–26].

Coupled to the collection of sediments, the acquisition of historical information retained in private and public archives is fundamental. It can help identify: (1) where specific point sources were located in the past; (2) if and how productions have changed through time; (3) which areas sustained substantial human-induced reworking of bottom sediments [27,28].

The number of sampling sites should be chosen in accordance with the purpose of the research: higher for a general site characterization of surficial sediments to check sources, distribution, and level of pollution; lower for more specific questions, such as sediment accumulation rates and dates, chronologies, fluxes, processes, and trends.

In this paper, results of the scientific approaches designed and applied for the study of environmental pollution issues in two Italian industrial areas are summarized. The locations are the Augusta Harbor in Southern Italy and the Venice Lagoon with its nearby industrial area of Porto Marghera in northeastern Italy. In both cases, the coupling of scientific evidence from sediment cores with historical information managed to provide a clear view of polluting sources, timings, and mechanisms.. Results were either debated in courts (Venice Lagoon) or used in the framework of preliminary investigations (Augusta Harbor). They represent perfect examples of how sediment records can be used as tools for environmental forensic studies. Scientific data have been published in specific papers on international journals [27,28], and this article aims to underline the study paths undertaken to answer the scientific-legal questions that have arisen during the investigations, sometimes not foreseen and often quite challenging. The acquired experience can be put at the service of developing countries that present similar risks of uncontrolled industrialization.

## 2. Study Areas and Historical Information

### 2.1. The Augusta Harbor

The Augusta Bay (Figure 1a) is a semi-enclosed natural basin occupying around 30 km of the eastern coast of Sicily (Ionian Sea, southern Italy) and covering an area of approximately 4000 ha [29,30]. The Augusta Harbor is the northern sector of the bay and was sheltered in the early 1960s with the construction of artificial breakwaters. Since then, the harbor is connected to the bay and the open sea only through two artificial inlets, called Scirocco (in the south, 300 m wide and 13 m deep) and Levante (in the east, 400 m wide and 40 m deep), respectively [30]. The Augusta Harbor is about 24 km$^2$ and the average water depth is approximately 15 m, with the maximum depth (40 m) in correspondence of the Levante inlet [31]. Exchanges with the open sea are mainly driven by tidal fluctuations and are consequently correlated with the entry/exit of tidal flows and relative amplitudes. Open sea waters enter predominantly from the Levante inlet (mean speed of 18 cm s$^{-1}$), then flow northward, parallel to the dam, slow and constantly with their pace and exit at the Scirocco inlet at a speed of 5–6 cm s$^{-1}$ [32]. The north-western sector of the harbor is locally influenced by the poor seasonal outflows of the Mulinello, Marcellino, and Cantera streams but their contribution is generally poor, with localized hydrodynamic effects [31]. Due to its confinement, the hydrodynamic regime of the Augusta Harbor is scarce, favoring the accumulation of nutrients and pollutants in sediments and the occurrence of eutrophication episodes [31,33].

The Augusta Harbor hosts one of the most important ports of the Mediterranean Sea, and many industries are located along its western part, including petroleum and petrochemical plants. The industrial development started in the 1950s with the construction of oil refineries and proceeded in the 1960s with the onset of a Mercury Cell Chlor-Alkali Plant (MCCAP), almost contemporary to the construction of the outer breakwaters [34]. Sediment dredging, disposal, and partial nourishment of the western coast have further modified the pristine environment of the bay. Indeed, 3/5 of the port surface (approximately 14.4 km$^2$) have been dredged so far while the dredged material might have been released over a large area of the bay on the way to offshore dumping sites (yellow areas; Figure 1a) [27].

Since the 1970s the area has become internationally recognized as a polluted environment from industrial and petrochemical plants, but also from agriculture and urban wastes, with consequent risks to the ecosystem and human health [30,34,35]. Specifically, mercury contamination from the MCCAP in the southern sector of the Augusta Harbor (Figure 1a) has produced sediment concentrations beyond international levels in parts of the area and is by far the most important environmental issue [27,34,36]. It is estimated that ~500 t of Hg have been discharged to the Augusta Harbor [30,34]. In the 1980s the demercurization and waste treatment plants started working in response to the Italian anti-pollution law 319 of 1976 (aka "Legge Merli", in Italian), and discharged treated wastewater outside the harbor. Owing to the high state of environmental degradation, this area was included in the 2002 National Remediation Plan by the Italian Environmental Ministry, then the MCCAP closed in 2005 with the removal of the Hg cells [32,34].

In the early 2000s, scientific evidence became necessary to support legal investigations against the industries that caused the pollution and who could potentially be expected to contribute to remediation costs. In 2003, researchers from CNR-ISMAR Bologna started work to provide such evidence.

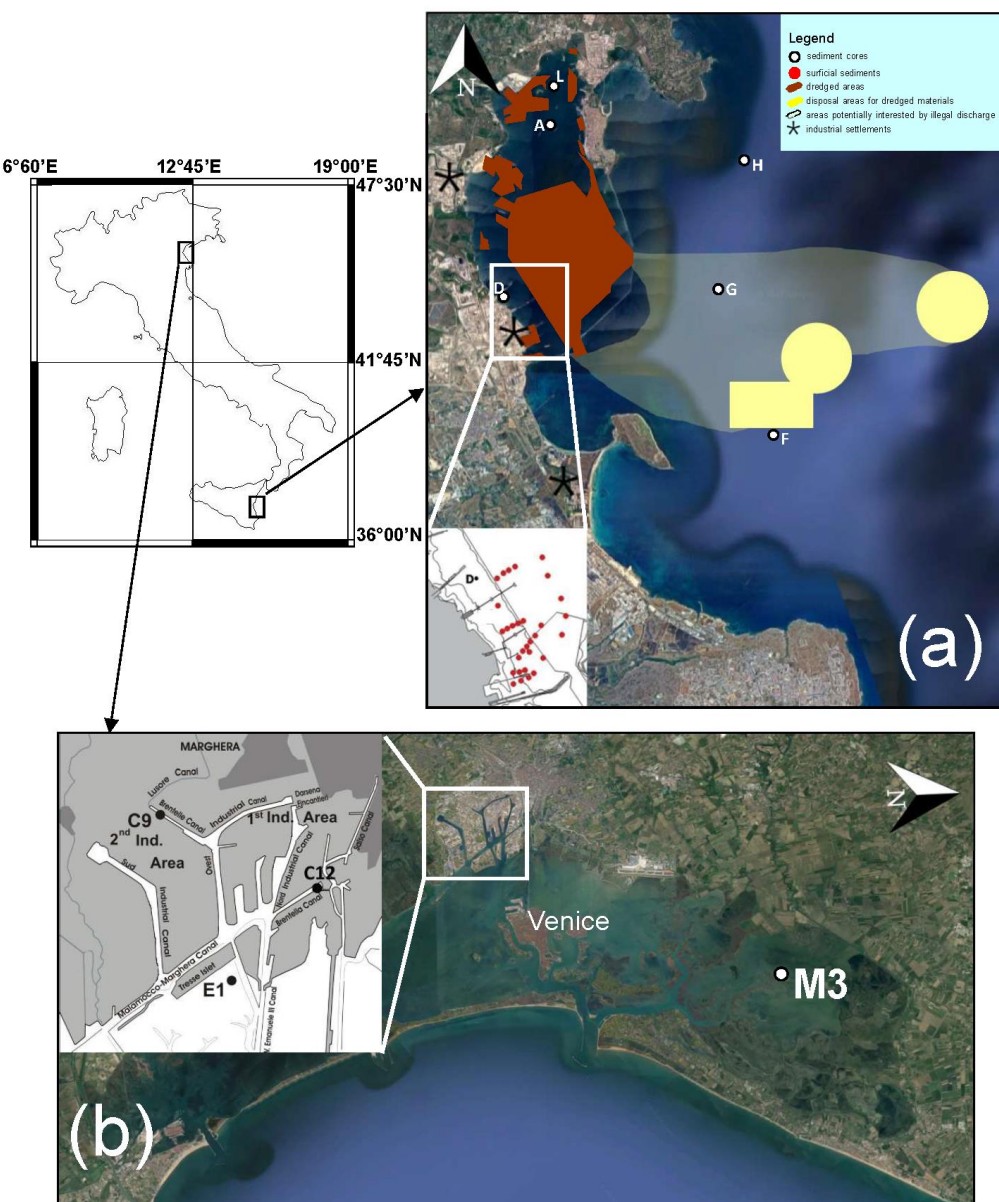

**Figure 1.** Location of study areas and sampling sites: (**a**) Augusta Bay with sampling sites, dredged and disposal areas, and locations of industrial settlements; (**b**) Venice Lagoon, with a snapshot over the Industrial Area of Porto Marghera. Satellite images are taken from Google maps.

### *2.2. The Venice Lagoon*

The Venice Lagoon (Figure 1b) is part of the Adriatic Sea in northeastern Italy. It has a surface area of 550 km$^2$, an average depth of 0.6 m, and a salinity of about 25–36‰ [37]. The water exchange with the Adriatic Sea (1.6–5.2 × 10$^8$ m$^3$ day$^{-1}$) occurs through tidal cycles at three entrance channels that divide the lagoon into three basins (Lido, Malamocco, and Chioggia), while freshwater (30.7 m$^3$ s$^{-1}$) enters the lagoon through about 24 tributaries [37,38]. The morphology is characterized by a network of canals of various depths, mud flats, tidal marshes, and islets that were first created during the maximum Holocene transgression 6000 years BP [39]. The sediment deposition rate in the lagoon is quite variable, and is higher in the northern sector, due to the supply from the Lido inlet [40,41]. Bottom sediments are composed almost exclusively of very fine sand and silt. Clay contribution is higher in the inner part of the lagoon and reflects the water dynamic: maximum at the inlets and within the main canals, and lower close to the mainland [38].

The Venice Lagoon is a unique shallow aquatic ecosystem whose complex morphology has been profoundly modified over time by human interventions. In particular, during the last century it has experienced a general degradation with the deepening of tidal flats (present rates of sediment loss are estimated to be 0.8 Mm$^3$ y$^{-1}$) and the reduction of salt marsh areas (from 168 km$^2$ in 1930 to 60 km$^2$ in 2002) [39,42]. In addition, the development of economic activities on the mainland have led to further changes: (1) the construction of the 1st Industrial Area of Porto Marghera at the beginning of the 20th century; (2) the excavation of the navigation canal Vittorio Emanuele III in 1930; (3) the building of the trans-lagoon bridge in 1934; (4) the development of the 2nd Industrial Area after World War 2; (5) the excavation of the navigation canal Malamocco Marghera, in 1969 [28]. More recently, the construction of the controversial *Modulo Sperimentale Elettromeccanico* (MOSE, Italian for Experimental Electromechanical Module) began in 2003 at the lagoon's inlets, with the aim of protecting the city of Venice from high tides (*Acqua alta* in Italian) through isolation from the adjacent Adriatic Sea [43] and is now fully operative.

Until the 1970s, Porto Marghera was one of the most important industrial areas in Italy, with plants for the production of chloro-alkali, sulfuric acid, ammonia, fibers, fertilizers from phosphorite, glass, coke, paints, detergents, firebricks, Al from bauxite, and the recovery of Zn and Cd from sphalerites [6,28,38]. The contamination of lagoon waters and sediments began soon after the first economic development, when industrial, urban, and agricultural discharges were disposed freely into the lagoon [41]. The situation became so serious that, at the end of the 1990s, a trial against managers of the petrochemical plants began. The defendants were charged with pollution and environmental disaster by polychlorodiphenyl dioxins and furans (PCDD/Fs) in the industrial area of Porto Marghera and in the whole Venice Lagoon. The main subject of the lawsuit was the production of vinyl-chloride monomer (VCM) that, at that time, was considered the principal industrial source of PCDD/Fs in this region. Researchers from CNR-ISMAR Bologna were involved for their expertise in order to provide scientific evidence in support or against this hypothesis, and the results of their studies served in the court hearing as a technical report.

## 3. Materials and Methods

### 3.1. Research Activity in the Augusta Bay Performed by ISMAR-CNR Bologna

A seismic survey in the Augusta Bay was carried out in October 2003 using a CHIRP II (Datasonic) sub-bottom profiler with the following operational characteristics: frequency sweep between 2 and 7 kHz; sampling interval of 125 ms; high resolution of a few centimeters; SeisPrho processing software [27,44]. Internal structures of the sediment layers were recognized and permitted to discriminate between undisturbed, laminated sediment layers, and chaotic sequences (Figure 2a). Bottom reflectivities were calculated in order to obtain a first approximation of grain size structures [44].

The sampling strategy for the definition of Hg contamination in sediment cores was designed according to seismic data, bottom reflectivities, and historical informationvi retrieved from local public (Port Authority, Municipality of Siracusa) and private (industries) archives relative to the dredging and dumping areas. As a result, much attention was paid during sampling to avoid the dredged and dumping areas or zones close to piers and/or influenced by docking and manoeuvring of ships. Core G (Figure 1a) was sampled on purpose in an area of supposed illegal dumping of dredged material from the harbor, in order to check the existence and effects of this practice. Sediment cores were collected with the gravity corer SW-104, patented by CNR and especially designed to preserve undisturbed sediment–water interfaces.

Newly arisen legal questions relative to the Hg sources in surficial sediments asked for a second sampling in 2005. Surficial sediments were sampled in the southern sector of the Augusta Harbor (Figure 1a) in order to be analysed for Hg and hexachlorobenzene (HCB) [27].

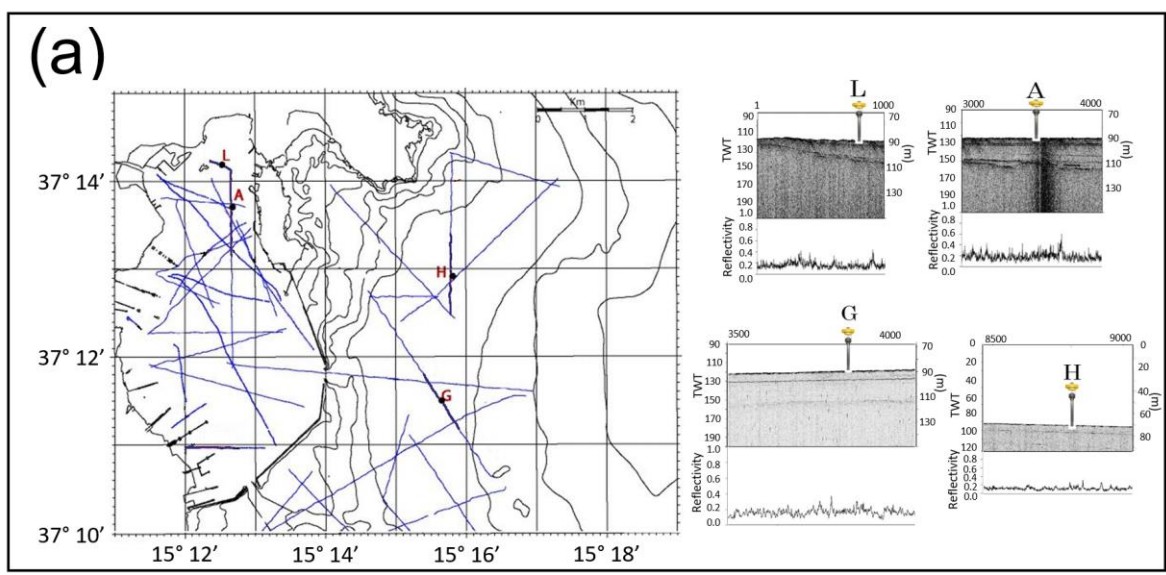

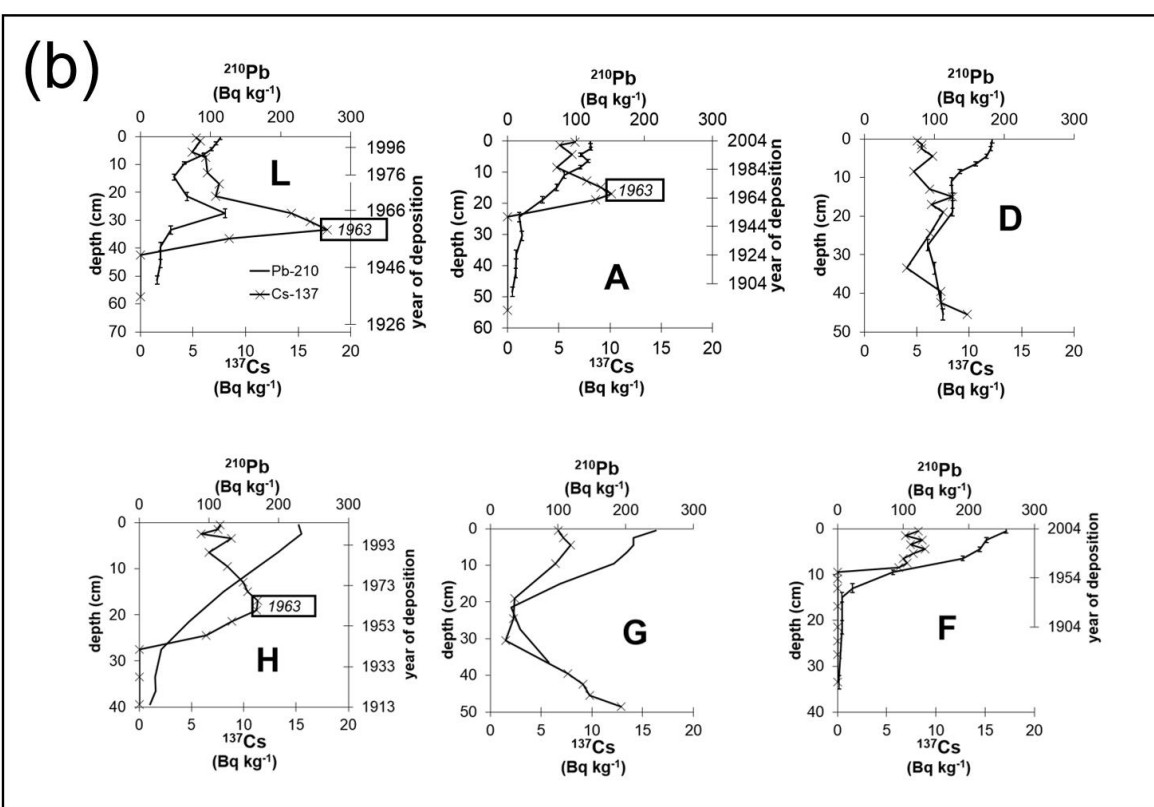

**Figure 2.** (**a**) Map of the seismic survey realized in the Augusta Bay and Harbor, together with CHIRP profiles and seismic surface reflectivities relative to L, A, H, and G core locations; (**b**) $^{210}$Pb and $^{137}$Cs activity−depth distributions for cores L, A, D, H, G, and F. Profiles are presented also vs. years of deposition, calculated from the S values. When detected, the $^{137}$Cs 1963 peak is indicated. Modified from [27].

After collection, sediment cores were kept vertical (in order to avoid intra-layer mixing and intestitial water migration) while scanning for magnetic susceptibility (using a Bartington meter, on each centimeter every 10 s) and X-radiographies (with a radiogenic industrial tube Gilardoni CPX-91 160 equipped with a Be window and industrial camera films). These two parameters are useful for a first-rate approximation of internal structures

and as a support for chronologies. Indeed, if highly magnetic tephra layers are observed, then a correlation with historical eruptions can be made. Cores were then extruded and carefully sectioned in 1–3 cm intervals with higher resolution at the top. Sediments were analysed for bulk dry density (assuming a particle density of 2.5 g cm$^{-3}$), organic matter content (% of C and N, $\delta^{13}$C and $\delta^{15}$N on a Carlo Erba CHNS Elemental Analyser), mineral composition (from 2° to 60° on a Philips PW 1710 CuK X-ray diffractometer, operating at 30 mA and 40 kV), grain size (through wet sieving and analyses with an X-ray Micromeritic SediGraph set on a 4–12 $\phi$ size range) and chronology. This latter was obtained through: (1) acid extraction and alpha spectrometry of $^{210}$Po for $^{210}$Pb determinations (assuming secular equilibrium), and (2) non-destructive gamma spectrometry of dry samples for $^{137}$Cs. For $^{210}$Pb, accuracy was estimated by repeated measurements of the standard reference material (SRM) IAEA 300 (Baltic Sea sediment) and the results were within uncertainties. Precision, calculated from independent analyses of the same sample, was 4.6%. For $^{137}$Cs, accuracy was checked by counting the international SRM S7 IAEA Baltic Sea sediment, and the results were within uncertainties. Precision, estimated by repeated analyses of the same sample, ranged between 2.05 and 3.07%. Radionuclide profiles for cores L, A, D, H, G, and F are reported in Figure 2b.

Hg was extracted following the procedure outlined in the EPA 3051A REV.1 (https://www.epa.gov/sites/default/files/2015-12/documents/3051a.pdf, accessed on 6 June 2023), including drying at 40 °C, acid microwave extraction, centrifugation, and evaporation. Concentrations were determined following the method EPA 6010 C, by use of an inductively coupled plasma-atomic emission spectrometer (ICP-AES) Spectro Ciros. Reproducibility, obtained with repeated measurements of the same sample, was 8.2%. Recovery, defined with the SRM MESS-1, was 81%. The analytical detection limit was 0.1 mg kg$^{-1}$.

For HCB analyses, dried samples were spiked with $^{13}$C$_{12}$-HCB (CLM351; Cambridge Isotope Laboratories, Woburn, MA, USA) as internal standard, extracted by ASE 200 (DIONEX, Sunnyvale, CA, USA; with an n-hexane/dichloromethane solution), and cleaned up using the automatic system, Dioxin Prep (Fluid Management System Inc., Lexington, KY, USA), as defined by method EPA 8270 (https://19january2017snapshot.epa.gov/sites/production/files/2015-07/documents/epa-8270d.pdf, accessed on 6 June 2023). High resolution gas chromatography mass spectrometry (HRGC-MS) analyses were performed on an HP 6890 plus gas chromatograph coupled to a Micromass Autospec Ultima mass spectrometer. The SRM solution for quantification was EC1668 (Cambridge Isotope Laboratories, Woburn, MA, USA). Recovery (calculated with three $^{13}$C$_{12}$-labeled PCBs from SRM EC4979) was 86%, while reproducibility was 2.1%. The analytical detection limit was 1 mg kg$^{-1}$.

Hg and HCB concentrations were calculated with respect to dry weight. For the purpose of this paper, only results of radionuclide dating and chemical measurements (Hg and HCB) are summarized. Details of the other measurements can be found in [27].

### 3.2. Research Activity in the Venice Lagoon and the Porto Marghera Industrial Area Performed by ISMAR-CNR Bologna

A series of published papers and documents issued by public authorities were reviewed in order to define the state of the art of scientific studies. References are reported in detail in [28,38,45–48].

Sediment cores considered in this paper provide a comprehensive view of the potential that can be expressed by sedimentary records involved in environmental forensic studies: (1) cores E1 and M3 represent a lagoon site near the Tresse Islet sampled in 1996 and a salt marsh of the Northern lagoon collected in 1998, respectively; (2) core C9 was collected in 1996 in the 2nd Industrial Area, and (3) core C12 was collected in the same year in the 1st Industrial Area (Figure 1b).

Sediment cores were analysed for chronology and ancillary parameters, with the same methods reported in Section 3.2. Analysis performances were similar to those reported for chronologies.

The analysis of PCDD/Fs on freeze dried sediments included the spiking with a series of 15 [13]C-labelled PCDD/F congeners as internal standards, then a Soxhlet-extraction with toluene, as determined by the EPA 3540 method (https://www.epa.gov/sites/production/files/2015-12/documents/3540c.pdf, accessed on 6 June 2023). Then the analysis continued through clean-up and detection procedures defined by the HRGC-MS EPA 1613/94/revision b method (https://www.epa.gov/sites/default/files/2015-08/documents/method_1613b_1994.pdf, accessed on 6 June 2023), for the determination of 17 2,3,7,8 substituted congeners with an HP 6890 Plus gas-chromatograph coupled to a FINNIGAN MAT 9SS mass spectrometer. Five SRM solution injections (EDF 9999 Cambridge Isotope Laboratories, Woburn, MA, USA) were used for quantitative determinations, and recoveries were always between 40% and 120% [45]. Toxic equivalent quantities (TEQs) were calculated using the International Toxic Equivalent Factors (I-TEFs) [49]. The composition of PCDD/F mixtures was calculated as the permil (‰) contribution of each homologue group with respect to the total.

Arsenic was analyzed on freeze dried samples by atomic absorption spectrometry (AAS) after total dissolution. Accuracy was within the uncertainties for SRM NIST 2710 (Montana Soil), and precision was 7% [28].

## 4. Results and Discussion

### 4.1. Assessment of Hg Delivery to the Augusta Harbor: Direct Inputs or Resuspension?

The chlor-alkali plants discharged without treatments in the harbor until the 1970s, when a demercurization plant and the consortial biological depurator for waste treatments became operative, in response to national laws. In spite of that, high Hg levels are still measured in surficial sediments [30,34–36] implying the possibility of a recent contribution. As written in Section 2.1, this possibility was the focal point to be verified by the historical and scientific reconstruction here summarized.

Sediment chronologies defined for the cores collected in 2003 in the Augusta Bay and Harbor were of great interest for the purpose of this scientific investigation. For most sites, they covered time intervals sufficiently extended to include the industrial setting, its development, and decline (i.e., early 1960s–late 1980s) up to 2003 (Figure 3a). $^{210}Pb_{ex}$ derived accumulation rates (S, cm y$^{-1}$) were calculated with the assumption of constant flux of the radiotracer and constant sedimentation (CF-CS), thus providing an average S useful for a first approximation dating, but probably overestimated. Associated errors for S ranged from 0.01 cm·year$^{-1}$ at the core top to 0.05 cm·year$^{-1}$ at the core bottom [27]. Since no evaluation of the effect of bioturbation could be achieved, surficial levels were excluded from its calculation. Despite these difficulties, in most cases $^{210}Pb$ derived chronologies were confirmed at deeper layers by $^{137}Cs$ results (see Figure 2b) and other parameters (e.g., magnetic susceptibility peaks, HCB production history) [27]. S values in the Augusta Bay ranged from 0.21 cm year$^{-1}$ (site F) to 0.42 cm year$^{-1}$ (site H), in agreement with box core data from the same area [50] and determined that 50 years of industrial history of the area were represented by 11–21 cm thick sediment layers. S values in the Augusta Harbor were higher (from 0.43 cm year$^{-1}$ at site A to 1.12 cm year$^{-1}$ in the sub-surficial sediment of site L) therefore the time intervals represented by these cores are shorter, starting from the mid-1940s or early 1950s.

As expected, site G proved to be very peculiar, with an unrealistic high rate of 1.08 cm·year$^{-1}$, most likely influenced by the massive sediment input generated by the undocumented illegal dumping of dredged material on the way to the disposal areas. Further investigation through core sampling at different locations throughout the Augusta Bay were necessary to verify the extent and pathways of this illegal habit (Figure 1a) but it was beyond the scope of the presented research. Chronologies were also confirmed by magnetic susceptibility measurements related to documented Etna eruptions in 1892 [51].

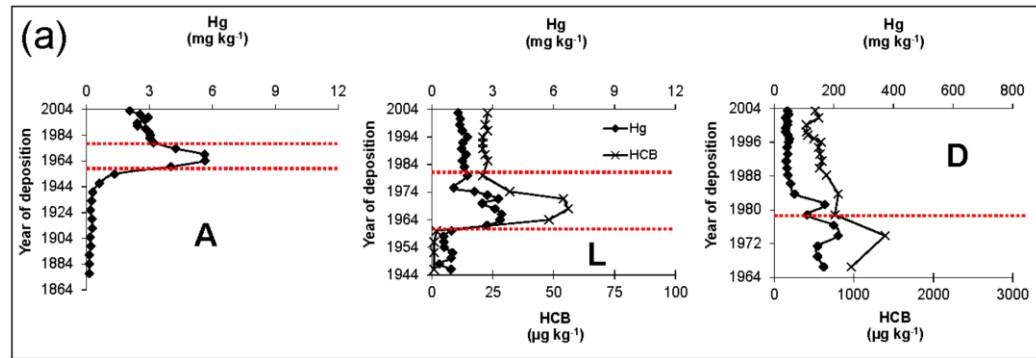

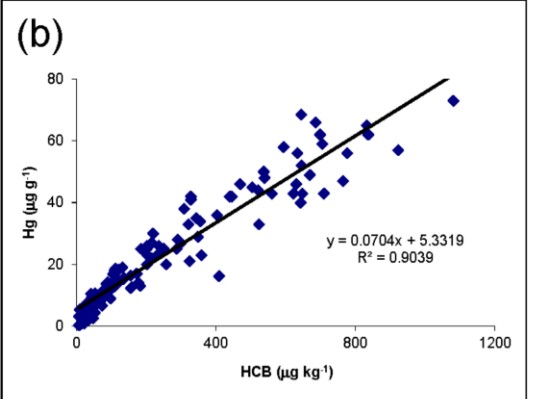

**Figure 3.** (**a**) Hg concentration vs. year of deposition in cores A, L, and D. HCB concentration profile for core L is also reported. The dotted lines encompass the period from the beginning of industrialization in the area (early 1960s) to the promotion of the first remediation measures (late 1970s); (**b**) Hg and HCB correlation in surficial samples collected in 2005 and in surficial layers of cores collected in 2003. The product moment and partial correlation coefficient R is 0.93, significant at $p < 0.0005$, and calculated with Statistica 7.0 StatSoft ©). Modified from [27].

Great variability was observed for Hg concentrations in sediments: values ranged from a minimum of 0.07 mg kg$^{-1}$ to a maximum of 575 mg kg$^{-1}$; [27], due to the presence of point sources that strongly influenced nearby locations. Chronologies were reliable enough to allow the reconstruction of Hg historical trends in sediments of the Augusta Harbor. As an example, Figure 3a shows Hg temporal profiles in cores A, L, and D (Figure 1a). The sedimentary record of core A covered the entire 20th century and evidenced significant variations of Hg concentrations, passing from near background values in the pre-industrial period to ca. 6–8 mg kg$^{-1}$ in the 1950s, at the peak of the industrial development of the area. Then, concentrations gradually and constantly decreased from the early 1970s, as the result of both technological improvements and remediation measures that came into use in that period [30–32,34–36]. However, the value measured at the sediment surface was still well above the natural background determined at the core bottom, as observed also in the box core sediments of the Augusta Bay [50]. In addition, such a surface value (0.25 mg kg$^{-1}$; Figure 3a) was still higher than the consensus-based threshold effect concentration of 0.18 mg kg$^{-1}$ reported by [52], posing a serious threat to biota and people. The same trends and events were recorded with more detail at site L, as the last 50–60 years of industrial history in the area had been represented by a higher number of measurements. This chronological reconstruction was impossible for core D, due to a highly disturbed sedimentary record, but its proximity to the MCCAP in the southern sector of the Augusta Harbor explained the very high Hg values at depth, still high even at the surface (92 mg kg$^{-1}$; Figure 3a).

These high Hg values in surface sediments demanded further investigations because they could have been determined either by: (1) the presence of active industrial sources or

(2) the deposition of resuspended old, contaminated sediment layers. The first option is a serious violation of current regulations and, if verified, would have justified legal actions against the last managers of the industrial plants. Indeed, the Italian anti-pollution law 319 of 1976 came into effect in 1979 and from that moment graphite anodes present in Hg cells of the chloralkali plant were removed. This happened because the demercurization plant could not work with the sludge deriving from the graphite anodes [53]. No active source of Hg in the Augusta Harbor could then be tolerated from that moment on.

The second option was equally critical as it assigns the responsibility for the contamination of surficial sediments primarily to dredging activities and their management by the authorities in charge.

The new scientific challenge was then to establish which of the two was the process at work in the Augusta Harbor. In order to provide a reliable answer, an alternative tracer of industrial pollution with a perfectly time-constrained history in the study area had to be found. After further historical research, the choice fell on HCBs, whose input chronologies in the Augusta Bay were well known and documented, as these compounds are formed by graphite anodes in chloralkali plants, and as by-product of the manufacture of tetrachloroethylene (TCE) [54]. In the industrial area of Augusta, TCE production started in 1959 and ceased definitively in 1979, taking the following nine years for complete decommissioning. In addition, production wastes were discharged in the Augusta Harbor with no treatment or disposal procedures until 1976. In the last 40 years, no HCB sources have been active in the area [54].

The HCB temporal record for core L (Figure 3a) was consistent with the timing of TCE production in the Augusta Bay until the 1980s; values were close to detection limits in the preindustrial period, then increased from the early 1960s, reached the maximum around the beginning of the 1970s, and then abruptly decreased at the end of the 1970s. However, if sediments were to deposit undisturbed from that moment on, measured values had to be close to zero, since HCB inputs ceased definitively in the 1980s. On the contrary, measured values in the topmost section remained quite constant around 25 $\mu$g kg$^{-1}$ (Figure 3a). The highly significant correlation observed between Hg and HCB in surficial sediment samples (Figure 3b) was interpreted as evidence that the two contaminants were controlled in surface sediments by the same depositional process. This was likely linked to the mobilization of polluted sediments deposited in the period 1958–1980 by dredging and/or ship traffic. This process can also explain the lower values of $^{210}$Pb observed in levels dated around the 1980s (Figure 2b), when dredging activities began [27]. For example, core L activities measured at 15 cm depth (dated with the CF-CS model around the 1980s; Figure 2b) are lower than those of the immediately deeper levels, suggesting the influence of relatively older sediments (and therefore with lower $^{210}$Pb activities). These sediments add to those naturally settling in that area and have the effect of lowering the detected total activity. No evidence was found that could prove the existence of direct inputs from productions, at least in the northern sector of the Augusta Harbor at the time of sampling. The second option was then held responsible for the high Hg values at the sediment surface. Consequently, management recommendations derived by this research included the adoption of all possible measures to control the re-suspension and re-deposition of sediments whenever dredging activities were necessary or during ship transit [55,56]. Indeed, other research studies in the following years focused on the possibility that Hg discharged in the Augusta Harbor might be a primary source to the adjacent marine system through the outflow of bottom waters along its steep slope and gullies [57] and a secondary source through reworking, resuspension, and transport [58]. In addition, ref. [57] also recognized that a substantial contribution to Hg contamination in the Augusta Bay might result from past uncontrolled dredging activities that deposited part of the contaminated sediments on the slope system outside the harbor (Figure 1a). Additional risks posed by the high seismicity of the area were measured with the modelling of the sudden breaching and complete collapse of the system damming by a destructive earthquake [59]. The resulting dispersion of contaminants (using Hg as an example of those present in the study area) is expected to occur almost instantaneously at a

large scale, suggesting that natural and artificial hazards are inextricably connected and demand innovative methodological approaches for planning appropriate risk reduction and policy management [59]. As the environmental persistence of Hg and HCB availability facilitates their bioaccumulation and affects the health status of marine organisms, the possible implications for environmental risk, pollutant transfer, and human health were investigated by other studies in the same years [35,60]. Specific analyses on Hg content and the Hg isotopic signatures of sediments, fish, and human hair shown as pelagic fish consumption is the principal transfer mechanism to the human population [35]. In addition, ref. [60] observed the progressive decrease of bioavailability and biological effects on local fauna over the last 20 years, is likely linked to the closure of many industrial facilities.

*4.2. The origin of PCDD/F Contamination in the Industrial Area of Porto Marghera: A Compelling Scientific Investigation*

The identification of specific industrial processes responsible for the presence of PCDD/Fs in the area of Porto Marghera (Venice) was the core of the trial against past and contemporary managers of the petrochemical plants during the late 1990s–early 2000s [6,28]. Research studies aiming at providing scientific evidences on this regard started from the observation of contaminant concentration distribution and the relative congener composition pattern (i.e., PCDD/F signatures or fingerprints) within industrial canals and lagoon environments. Two main PCDD/F fingerprints were identified (Figure 4a): (1) one characterized exclusively the core collected in the Lusore–Brentelle canal in the 2nd Industrial Area (core C9, Figure 1b), while (2) the other was found in the canals of the 1st Industrial Area (core C12 in the Brentella Canal, Figure 1b) and at sub-surficial depths in cores collected all over the lagoon (cores E1 and M3, Figure 1b). The first pattern was composed almost exclusively of octachlorodibenzofuran (OCDF) with no significant contribution by other furans and dioxins, while the second one was characterized by the presence also of other heavy furans (Figure 4a). In addition, samples collected in the 1st Industrial Area showed concentrations sensibly higher than those measured in the 2nd Industrial Area (core C9, Figure 1b). This first evidence testified the presence of two different sources of PCDD/Fs in the industrial canals of Porto Marghera, one in the 1st Industrial Area and another in the 2nd Industrial Area. Different types of industries have settled in the two industrial areas: chemical and electro-metallurgical industries were located mostly along the Nord Industrial Canal in the 1st Industrial Area, while the 2nd Industrial Area hosted petrochemical plants [61]. It was therefore legitimate to hypothesize that the two observed PCDD/F signatures were linked to these different industrial clusters. The questions arising from this first evidence were then as follows: (1) Were there in the 1st Industrial Area specific industrial metal processes that could have given levels of contamination and footprints of PCDD/Fs comparable to those measured in core C12 (Figure 4a)? and (2) Was it possible to recognize a similar link between industries located along the Lusore–Brentelle Canal (2nd Industrial Area) and the congener composition of dioxins and furans measured in its sediments (core C9; Figure 4a)? The results of the challenging literature search required to answer the first question are now summarized in the following part.

Ref. [62] tried to correlate the contamination of the Venice lagoon sediments to potential sources, analyzing literature data (i.e., contaminated emissions and sediments) through a statistical approach (principal component analysis, PCA). Lagoon sediments presented in [62] for the Venice urban area and at the boundary with Porto Marghera fell into the same group with the sediment of the Norwegian Frijefjorden contaminated by a magnesium processing plant [63]. This industrial process involved the chlorination of Mg under carbon presence at high temperatures (800–900 °C) and plant discharges were found to be enriched in OCDF, with PCDD/F emissions in water amounting to about 500 g TEQs per year in 1989 [63]. Indeed, the congener composition (as chlorination classes, Figure 4b) of the Norwegian fjord sample was characterized by the presence of heavy furans with a small contribution of octachlorodibenzodioxin (OCDD), a pattern similar to that observed in sediments of the 1st Industrial Area (C12, Figure 4a). In addition, total concentrations of

PCDD/Fs measured in the fjord and in the more contaminated canals of the 1st Industrial Area were of the same magnitude (hundreds of thousands of ng kg$^{-1}$, Figure 4a,b). A study conducted in 1997 [64] verified the level of sediment contamination about ten years after the emissions from the Mg production plant in the Frijefjorden had been drastically reduced. The results showed that pollutant levels in surface sediments were reduced by about two orders of magnitude (in the most contaminated area, where Mg processing slags were located, concentrations passed from about 37,000 to 3800 ng kg$^{-1}$ TEQs), but the PCDD/F fingerprints were unchanged. This proved incontestably that the Mg industry was the major factor responsible for PCDD/F pollution at that site, with negligible contribution from other production processes.

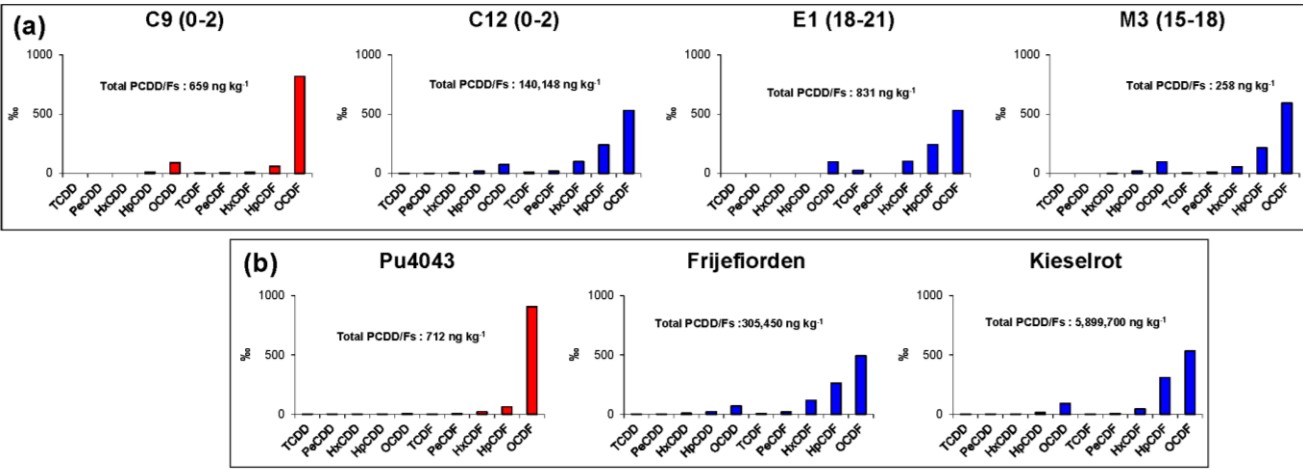

**Figure 4.** (**a**) PCDD/F congener composition (as chlorination classes) in the Venice Lagoon (E1 and M3) and in the industrial canals of Porto Marghera (C9 and C12). Total PCDD/F concentrations are also reported. Data are from [28,45,46]. (**b**) PCDD/F congener composition (as chlorination classes) retrieved in literature and relative to specific industrial processes. Total PCDD/F concentrations are also reported. Data are from [63,65,66].

As no Mg industry had ever been present in the Industrial Area of Porto Marghera, a new question arose: Was it possible to identify another industrial metal process that could explain the observed PCDD/F signatures in the 1st Industrial Area? At that point, it became necessary to verify whether the sediments of the most contaminated canals of Porto Marghera presented associations between PCDD/Fs and metals that could lead to specific industrial processes actually operating in the area. Using data from the survey carried out by the Venice Water Authority and the Port Authority [67], a good correlation ($R^2$: 0.742) was found between PCDD/Fs and As (Figure 5).

Since the formation of PCDD/Fs requires oxygen, a hydrocarbon compound, Cl, a catalyst (e.g., Cu or Fe), an ideal temperature (400–600 °C), and an adequate residence time [68], there was the necessity to identify the industrial processes operating under these conditions and simultaneously related to As pollution. One of these processes occurred regularly in the 1st Industrial Area and involved the extraction of Cu from pyrite ashes (very rich in As [69]), through the addition of 10% NaCl (chlorination) and the subsequent roasting at temperatures of about 450–550 °C with the formation of $CuCl_2$ and $FeCl_2$ [70]. Both compounds (and also NaCl) are regarded as catalysts for the formation of dioxins and especially furans [68,71], through chlorination and oxidation reactions of carbonaceous particles with degenerate graphitic structure, present in the combustion gases. Reactions take place on metal oxides in the flying ashes and are catalyzed by transition metal compounds. Chlorination takes place essentially by means of inorganic salts (KCl, CuCl, $CuCl_2$, NaCl) while the oxygen supply is given both by that present in the gaseous phase and by that existing in the aromatic nuclei present in the degenerate graphitic structures. Table 1 summarizes the main reactions that occur when roasting pyrite

ashes. It is evident that a high amount of metal compounds potentially active as PCDD/F catalysts is produced. The residues of the roasting are then leached and metallic Cu, small amounts of Ag, Au, and also Zn are recovered from the resulting solution. Furthermore, the residual Fe oxides (called "purple ore") are agglomerated into tiles through a sintering process that, due to the presence of residual chlorine from the previous treatment, can produce dioxins and furans.

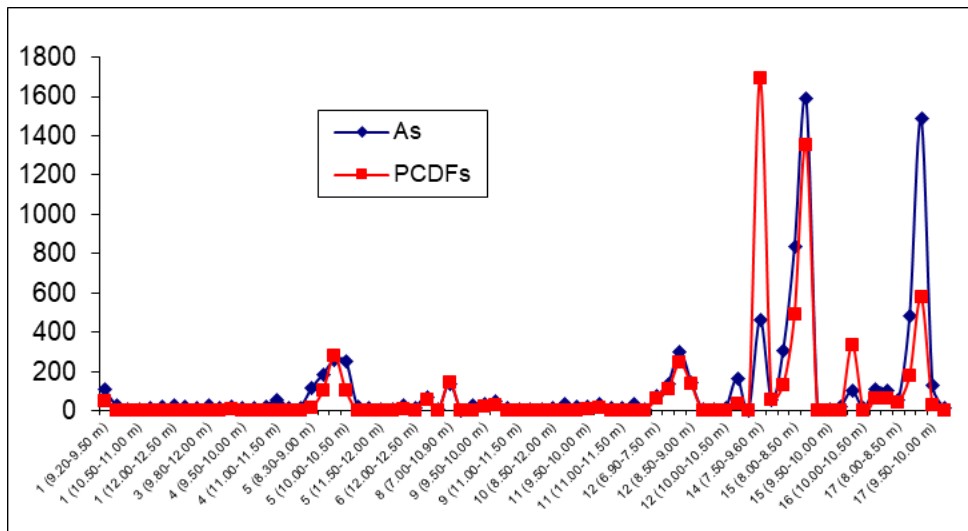

**Figure 5.** Association between As peaks ($\mu$g g$^{-1}$) and PCDD/Fs ($\mu$g kg$^{-1}$) in sediments of the industrial canals Nord and Brentella. Data are from [67].

**Table 1.** Main chemical reactions occurring during the roasting of pyrite ashes [67].

| |
|---|
| 1. Double exchange between iron sulfate or copper sulfate with sodium chloride<br>$FeSO_4 + NaCl \rightarrow Na_2SO_4 + FeCl_2,$<br>$CuSO_4 + NaCl \rightarrow Na_2SO_4 + CuCl_2,$ |
| 2. Oxidation of iron (II) chloride to iron (III) chloride<br>$12\ FeCl_2 + 3\ O_2 \rightarrow 8\ FeCl_3 + 2\ Fe_2O_3$ |
| 3. Reaction of iron (III) chloride with Cu sulfide<br>$4\ FeCl_3 + 3\ Cu_2S + 6\ O_2 \rightarrow 6\ CuCl_2 + 2\ Fe_2O_3 + 3\ SO_2$ |

The industrial process able to produce PCDD/Fs as by-product was identified, but the so-called smoking gun had still to be found, i.e., a case study that linked undoubtedly the process of Cu extraction from pyrite ashes to PCDD/F production. A new literature search brought to light the "Kieselrot" case [59]. Kieselrot (German for red pebble) is the name given to the red siliceous slag obtained from Cu extraction processes which were carried out in Marsberg (North Rhine-Westphalia, Germany), until 1945. The extraction process involved the chlorination, roasting, and leaching of a "black siliceous mineral" containing Cu called Schwarzkiesel. During the 1950s–60s, the Kieselrot was used to build sport fields, recreational areas, roads, and paths. In the following years, this material was found to be highly contaminated by chlorinated aromatic hydrocarbons, in particular PCDD/F, HCB, polychlorinated biphenyls, polycyclic aromatic hydrocarbons, as well as by heavy metals [65]. The concentration of PCDD/Fs measured in a Kieselrot slag was extremely high (5,899,700 ng kg$^{-1}$, Figure 4b), comparable to the highest values found in sediments of the Brentella Canal (Figure 5). The relative signatures were the same, with OCDF as the most abundant congener, followed by heavy furans and a small amount of OCDD (Figure 4a,b). In addition, As was measured in Kieselrot slags with a good correlation with PCDD/Fs [65], as observed also in the industrial canals Nord and Brentella of the 1st Industrial Area of Porto Marghera (Figure 5). No further evidence was needed, and the PCDD/F contamination found in the Brentella Canal was then ascribed entirely to sources

linked to Cu extraction from pyrite ashes. This industrial process had been active in the 1st Industrial Area since 1927 and, until the decommissioning in the early 1970s, produced about 130,000 units year of tiles and iron minerals [72].

The answer to the second question was more straightforward: the Lusore–Brentelle Canal was under the direct influence of a VCM stripping plant and the comparison of PCDD/F congener composition measured at site C9 with that observed in sediments from the Gulf of Finland under the direct influence of VCM stripping discharges (sample PU4043, [66], Figure 4b) left no doubt about the origin of dioxins and furans in the two areas. Therefore, it was concluded that VCM production in the 2nd Industrial Area affected only a restricted nearby area.

### 4.3. The History of PCDD/F Contamination in the Venice Lagoon

As already evidenced in Section 4.1, also in the Venice Lagoon the acquisition of reliable sediment chronologies was a mandatory prerequisite for the correct definition of PCDD/F temporal trends. Once sedimentary levels are correctly associated to a reasonable range of deposition years (i.e., an error confined below 5 years in the most recent levels, increasing to 20–30 years at higher depths) it is possible to "read" sediment profiles as historical documents.

Salt marshes are coastal wetlands that are regularly flooded and drained by seawater following the tidal cycle. They are found mostly at middle to high latitudes and are common habitats in the Venice Lagoon, where they are called "barene". Among their many ecosystem services [42], Venetian salt marshes represent an important historical and scientific archive that is able to record important environmental parameters over many decades [23,47,48], when their average accretion rates are comparable to the long-term average rate of sea level rise (SLR) in the lagoon (~0.25 cm year$^{-1}$; [23]).

Figure 6 shows the pattern of PCDD/F contamination recorded by salt marsh core M3 (Figure 1b). It is evident that PCDD/F concentrations reached the highest values between 1940 and 1980, and then decreased significantly. This $^{210}Pb_{ex}$ dating is highly reliable, since it was verified by Cs-137 peak activity at 4 cm depth, following the Chernobyl accident in 1986. The recent decreasing trend of PCDD/F concentrations in salt marshes was attributed to a strong reduction of inputs, due to the closure of the most polluting activities within the first industrial area, and to the promulgation of the first environmental regulations in Italy in the mid 1970s. In particular, the already cited Law 319/1976 regulated the composition of industrial effluents, and the Ministerial Decree 23 April 1998 ("Water Quality in the Venice Lagoon" published on the Official Gazette n. 140 and further revised in 1999) established very strict limits for industrial effluents to lagoon waters, banning the discharge of several toxic, persistent, and bioaccumulative contaminants (As, Cd, PCDD/Fs, PAHs, Hg, Pb, PCBs, chlorinated pesticides, cyanides, and tributyltin compounds, TBTs). During this period, several wastewater treatment plants became operative [28].

The definition of these contaminant input chronologies in salt marshes and lagoon sediments [73] has permitted assessment of the risks for both the environment and humans, with the use of models that were combined to create several past and future scenarios [74–76]. They have highlighted the predominant role of deep sediments as secondary sources of contamination through dredging [74,75] and the varying distribution pathways under different wind scenarios (i.e., bora and scirocco) [76]. Input chronologies were also used to implement a human intake and a physiologically-based pharmaco-kinetic model [77] used to simulate human internal exposure and provide a preliminary ecological and human health risk assessment for selected chemicals. Moreover, contaminant concentrations in sediments were at the base of a dynamic multimedia model applied to selected PCB and PCDD/F congeners, simulating the effects of climate change on their distribution and fluxes over the next 50 years in the Venice Lagoon (Italy) [78]. This model's results suggested that global warming would probably enhance PCB and PCDD/Fs mobility and hence their potential for long range atmospheric transport, reducing the environmental levels of these chemicals in the lagoon.

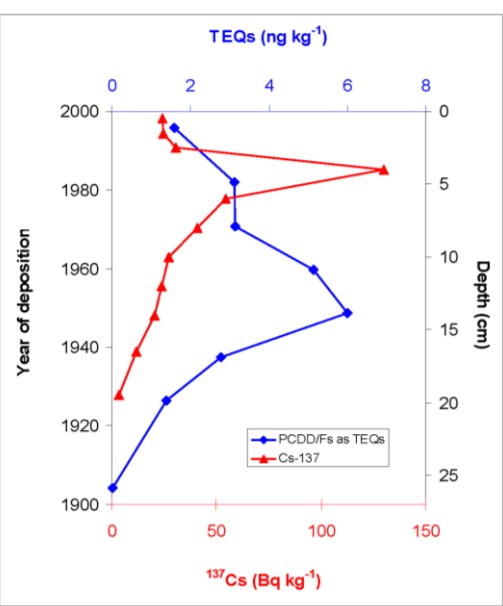

**Figure 6.** PCDD/Fs (as TEQs) in salt marsh core M3 reported vs. depth and year of deposition. The depth profile of $^{137}$Cs is also presented in order to evidence the reliability of the chronology (i.e., the peak activity of $^{137}$Cs corresponds to the fallout from the Chernobyl accident on 26 April 1986). Data are from [28].

## 5. Conclusions

The multifaceted approaches adopted for the presented research studies have proved to be suitable tools for obtaining a clear view of what happened, and why, in very complex study areas that were subject to important legal actions. The challenging environmental forensic studies requested on those occasions were relayed on sedimentary records and joined together information from field work (seismic, sampling of selected sites), laboratory analyses (sediment features and contaminants), and archive documents. In addition, a certain degree of flexibility and openness had to be exercised when unexpected questions arose in the course of the research/investigation: answers had to be found in places and methodologies that not always belonged to the scientific literature, such as public and private archives or technical books.

The case-histories here presented (Augusta Harbor and Venice Lagoon) happen to be located in the Mediterranean Sea but the described scientific approaches and investigation strategies can be applied all over the world, where similar problems of health security and industrial management might present in the next future. The environmental problems evidenced in the framework of these research studies were common in Western world developed countries as a result of unregulated and uncontrolled industrial development after World War 2. Nowadays, the same problems have a high probability of also affecting developing countries where intense industrialization and economic growth are increasing at a high pace but environmental regulations on sediment management are still incomplete or lacking. Lessons learned from these environmental forensic studies based on sedimentary records might be useful again. In addition, data provided by these studies were fundamental for implementing models that simulated the fate and distribution of contaminants and human exposure under different scenarios, including those of global change.

**Author Contributions:** L.G.B.: Conceptualization, methodology, original draft preparation, reviewing and editing. S.G.: Writing—reviewing and editing. All authors have read and agreed to the published version of the manuscript.

**Funding:** Funds for the research studies described were provided by Syndial S.p.A., Enichem S.p.A., and projects "Sistema Lagunare Veneziano" and "Orizzonte 2023".

**Institutional Review Board Statement:** Not applicable.

**Informed Consent Statement:** Not applicable.

**Data Availability Statement:** No new data were created or analyzed in this study. Data sharing is not applicable to this article.

**Acknowledgments:** The authors wish to thank M. Frignani, S. Albertazzi, S. Romano, E. Bacci, F. Fichera, G. Marozzi, F. Colombo, C. Carraro, and S. Racanelli for their valuable assistance. This is contribution n. 2077 by ISMAR-CNR of Bologna.

**Conflicts of Interest:** The authors declare no conflict of interest.

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
