# Peer review of "The Role of Sediment Records in Environmental Forensic Studies: Two Examples from Italy of Research Approaches Developed to Address Responsibilities and Management Options"

_applsci, doi:10.3390/app13126999_

Round 1
Reviewer 1 Report
Dear Authors,
I had the opportunity to review your manuscript " The role of sediment records in environmental forensic studies: two examples from Italy of research approaches developed to address responsibilities and management options."
GENERAL COMMENTS
I appreciated the idea of your review manuscript to combine the scientific evidence provided by sediment cores with historical information. I agreed with your view that this approach can be an effective way to reconstruct and quantify recent anthropogenic impacts in transitional and marine/coastal areas. Both are key points in studies aimed at establishing responsibility for environmental pollution.
DETAILED COMMENTS
The Introduction section is well organized and provides relevant bibliographical references to which I would add some papers that refer to different contexts, however relevant to the topic and very recent, such as:
-El Ouaty O., El M'rini A., Nachite D., Marrocchino E., Marin E., Rodella I. 2022. Assessment of the heavy metal sources and concentrations in the Nador Lagoon sediment, Northeast-Morocco, Ocean & Coastal Management, 216, 105900, https://doi.org/10.1016/j.ocecoaman.2021.105900.
-Xian H., Dong X., Wang Y., Li Yan, Xing J., Jeppesen E. 2022. Geochemical baseline establishment and pollution assessment of heavy metals in the largest coastal lagoon (Pinqing Lagoon) in China mainland. Marine Pollution Bulletin, 177, 113459, https://doi.org/10.1016/j.marpolbul.2022.113459
-Shetaia S.A., Abu Khatita A.M., Abdelhafez N.A., Shaker I.M., El Kafrawy S.B. 2022. Human-induced sediment degradation of Burullus lagoon, Nile Delta, Egypt: Heavy metals pollution status and potential ecological risk. Marine Pollution Bulletin, 178, 113566, https://doi.org/10.1016/j.marpolbul.2022.113566.
-Patra A., Das S., Mandal A., Mondal N.S., Kole D., Dutta P., Ghosh A.R. 2023 Seasonal variation of physicochemical parameters and heavy metal concentration in water and bottom sediment at harboring areas of Digha coast, West Bengal, India. Regional Studies in Marine Science, 62, 102945, https://doi.org/10.1016/j.rsma.2023.102945.
- Ebeid M.H., Ibrahim M.I.A., Abo Elkhair E.M., Mohamed L.A., Halim A.A., Shaban K.S., Fahmy M. 2022. The modified Canadian water index with other sediment models for assessment of sediments from two harbours on the Egyptian Mediterranean coast. Journal of Hazardous Materials Advances, 8, 100180, https://doi.org/10.1016/j.hazadv.2022.100180.
The description and explanation of the “study areas and historical information” and “materials and methods” sections are good. The study areas are well described. However, the quality of Fig. 1 and Fig. 2 could be improved.
Line 222 please check 210Po, maybe authors intended to write 210Pb
In the "Results and discussion" section the authors discuss in detail the results obtained from their study. I appreciated their use of schemes and figures. I suggest improving the quality of Figg. 3 and 4, if possible.
Author Response
Dear Authors
I had the opportunity to review your manuscript “The role of sediment records in environmental forensic studies: two examples from Italy of research approaches developed to address responsibilities and management options”
General comments
I appreciated the idea of your review manuscript to combine the scientific evidence provided by sediment cores with historical information. I agreed with your view that this approach can be an effective way to reconstruct and quantify recent anthropogenic impacts in transitional and marine/coastal areas. Both are key points in studies aimed at establishing responsibility for environmental pollution.
We thank the reviewer for his/her appreciation.
Detailed comments
The Introduction section is well organized and provides relevant bibliographical references to which I would add some papers that refer to different contexts, however relevant to the topic and very recent, such as:
- El Ouaty O., El M’rini A., Nachite D., Marrocchino E., Marin E., Rodella I., 2022. Assessment of the heavy metal sources and concentration in the Nador Lagoon sediment, North-east Morocco. Ocean & Coastal Management, 216, 105900, https://doi.org/10.1016/j.ocecoaman.2021.105900.
- Xian H., Dong X., Wang Y., Li Yan, Xing j., Jeppesen E., 2022. Geochemical baseline establishment and pollution assessment of heavy metals in the largest coastal lagoon (Pinqing Lagoon) in China mainland. Marine Pollution Bulletin, 177, 113459. https://doi.org/10.1016/j.marpolbul.2022.113459.
- Shetaia S.A., Abu Khatita A.M., Abdelhafez N.A., Shaker I.M:, El Kafrawy S.B., 2022. Human-induced sediment degradation of Burullus Lagoon, Nile Delta, Egypt. Heavy metals pollution status and potential ecological risk. Marine Pollution Bulletin, 178, 113566. https://doi.org/10.1016/j.marpolbul.2022.113566.
- Patra A., Das S., Mandal A., Mondal N.S., Kole D., Dutta P., Ghosh A.R., 2023. Seasonal variation of physicochemical parameters and heavy metal concentration in water and bottom sedimenta t harboring areas of Digha coast, West Bengal, India. Regional Studies in Marine Science, 62, 102945. https://doi.org/10.1016/j.rsma.2023.102945.
- Ebeid M.H., Ibrahim M.I.A., Abo Elkhar E.M., Mohamed L.A., Halim A.A., Shaban K.S., Fahmy M., 2022. The modified Canadian water index with other sediment models for assessment of sediments from two harbours on the Egyptian Mediterranean coast. Journal of Hazardous Materials Advances, 8, 100180. https://doi.org/10.1016/j.hazadv.2022.100180.
We thank the reviewer for his/her comment and added the suggested papers as reference in the manuscript and in the reference list.
The description and explanation of the “study area and historical information” and “materials and methods” section are good. The study areas are well described. However the quality of Fig. 1 and Fig. 2 could be improved.
The reviewer is right, therefore we have provided higher quality figures as .pdf files alongside the revised manuscript. See relative answer to Reviewer #3.
Line 222 please check 210Po, maybe authors intended to write 210Pb.
There is no mistake there: what is determined through alpha spectrometry is the activity of 210Po at 5.304 MeV that is then referred to as 210Pb because we assume that the two father/daughter radionuclides are in secular equilibrium (i.e. they have the same activity).
In the “Results and discussion” section the authors discuss in detail the results obtained from their study. I appreciated their use of schemes and figures. I suggest improving the quality of Fig. 3 and 4 if possible.
We thank the reviewer for his/her appreciation. We have provided higher quality figures as .pdf files alongside the revised manuscript, as also reported in a previous answer. See relative answer to Reviewer #3.
Reviewer 2 Report
the following questions need to be attended to by the authors:
(1) why did the samples used in this research for possible publication in 2023 collected between 1996 and 2005? this is about 27 years ago to 13 years ago.
(2) does it mean that for over two and half decades after collecting the samples that they are still valid and nothing has changed from the sampling site?
(3) how valid are they in 2023?
(4) why did it take about 9 years to collect the samples (between 1996 when the first sample was collected to 2005 when the last sample was collected)?
(5) the conclusion of the work is not in tandem with the abstract. the conclusion appeared so broad and did not address specifically the focus of the title of the research.
(6) did the conclusion address the study paths undertaken to answer challenging scientific-legal questions that might arise in such situations? if yes, it was not addressed
(7) could the answer to (6) above be drawn from a developing country? if yes, the research didnt project such
other comments are found on the attached manuscript

english language needs minor editing
Author Response
The following question need to be attended by the authors:
- Why did the samples used in this research for possible publication in 2023 collected between 1996 and 2005? This is about 27 years ago to 13 years ago.
The samples used in these researches were collected to answer specific scientific questions that emerged in the course of legal actions aimed at the correct attribution of management responsibilities for the pollution of the studied environments. These actions took place in the 1990’s and early 2000’s and the results were used during either Court debates and/or preliminary investigations. Scientific results were then published some years later (mid 2000’s – early 2010’s). In recent years, we have realized the need to give proper light to the study paths undertaken in those occasions (that were not fully described in the already published papers), mainly because the experience acquired could help developing similar approaches in countries that run the risk of uncontrolled anthropogenic impacts on the environment. For this reason, we think that the period elapsed from sample collection is not relevant, as the “modus operandi” described in this manuscript might still be useful for other studies.
- Does it mean that for over two and a half decades after collecting the samples that they are still valid and nothing has changed from the sampling site?
Environmental conditions in the studied areas have changed, and the results of the presented approaches have contributed in defining responsibilities and identifying the most suited remediation measures. However, the lesson learned from those two study cases is still valid and can be addressed to other areas in the world.
- How valid are they in 2023?
Their validity lays in the example they are of successful study approaches that have merged scientific evidences with historical reconstructions in order to reconstruct anthropogenic impacts to the environment.
- Why did it take out about 9 years to collect the samples (between 1996 when the first sample was collected to 2005 when the last sample was collected)?
The timing for sample collection and analysis was dictated by the requests arising from legal actions. This explains the temporal gap between the two samplings.
- The conclusion of the work is not in tandem with the abstract. The conclusion appeared so broad and did not address specifically the focus of the title of the research.
We understand that the reviewer would have preferred that the Conclusion section match and somehow repeat what written in the Abstract. However we preferred to devote part of the Conclusion section to highlight the potential of these approaches to be reproduced in similar context in other parts of the world, because that is the most important aspect of our contribution and we want it to be clearly stated.
- Did the conclusion address the study paths undertaken to answer challenging scientific-legal questions that might arise in such situations? If yes, it was not addressed
Definitely yes. We modified the manuscript accordingly to better clarify this crucial point. We might say that this is one of the main purposes of this paper.
- Could the answer to (6) above be drawn from a developing country? If yes, the research didn’t project such
As written in the previous answer, the manuscript was modified to better evidence this possibility.
Other comments are found on the attached manuscript
In the following section we answer to the comments that were visible in the pdf. file provided by the Editor.
Authors should be able to state clearly the methodology used. or may state what was determined according to the method of Bellucci et al Bellucci et al [22]
The details of the methodology for radiochronology, Hg and HCB analytical measurements are reported in the paper. Readers should refer to the cited reference if they wish to have a closer look to analytical results, since the main objective of the present manuscript is not the thorough discussion of them.
I wonder how samples collected in these area over 2 and half decades ago are used for research to draw conclusions now. Does it mean that nothing has changed for over 25 years ago?
As for the answer provided in question (2), we are aware that environmental conditions in the studied areas have likely changed through time, but the objective of this paper was to present approaches that have contributed in defining responsibilities and identifying the most suited remediation measures for impacted environments. For this reason, the lesson learned from those two study cases is still valid and can be addressed to other areas in the world.
Delete
We preferred to maintain this sentence because it clarifies that the cited references contain all the details about the documents that were reviewed during the research.
Quality of English Language
English language needs minor editing
The manuscript has been checked by a native English colleague and the language has been cleared by typos and bad constructed sentences. See also the relative answer to Reviewer #3
Reviewer 3 Report
The authors investigated "The role of sediment records in environmental forensic studies: two examples from Italy of research approaches developed to address responsibilities and management option.
Please check the English.
Please check and identify the novelty of work.
Some of figureas need to correct and improve.
The English needs to check.
Author Response
The authors investigated “The role of sediment records in environmental forensic studies: two examples from Italy of research approaches developed to address responsibilities and management options”
General comments
Please check the English
The manuscript has been checked by a native English colleague and the language has been cleared by typos and bad constructed sentences. See also the relative answer to Reviewer #2.
Please check and identify the novelty of work
The novelty of the work lies in its effort to provide indications on how very important legal questions have been answered by scientific research when management responsibilities had to be identified in case of environmental damage. We think that the examples presented can inspire/guide similar approaches by other scientists all over the world who are invited to provide scientific backgrounds to legal issues.
Some figures need to correct and improve
The reviewer is right, therefore we have provided higher quality figures as .pdf files alongside the revised manuscript. See relative answer to Reviewer #1.
Quality of English Language
The English needs to check
The manuscript has been checked by a native English colleague and the language has been cleared by typos and bad constructed sentences. See also the relative answer to Reviewer #2.
Round 2
Reviewer 3 Report
I must check the revised version based on highlights and reply to the answers. But can not see. So my decision is "Reject".
Author Response
I must check the revised version based on highlights and reply to the answers. But can not see. So my decision is "Reject".
Please find attached the revised manuscript with changes highlighted.
Below are the answers to the questions raised by the reviewer
Author's Notes
The authors investigated “The role of sediment records in environmental forensic studies: two examples from Italy of research approaches developed to address responsibilities and management options”
General comments
Please check the English
The manuscript has been checked by a native English colleague and the language has been cleared by typos and bad constructed sentences.
Please check and identify the novelty of work
The novelty of the work lies in its effort to provide indications on how very important legal questions have been answered by scientific research when management responsibilities had to be identified in case of environmental damage. We think that the examples presented can inspire/guide similar approaches by other scientists all over the world who are invited to provide scientific backgrounds to legal issues.
Some figures need to correct and improve
The reviewer is right, therefore we have provided higher quality figures as .pdf files alongside the revised manuscript.
Quality of English Language
The English needs to check
The manuscript has been checked by a native English colleague and the language has been cleared by typos and bad constructed sentences.

Round 3
Reviewer 3 Report
The authors investigated "The role of sediment records in environmental forensic studies: two examples from Italy of research approaches developed to address responsibilities and management options". In the cover letter, they did not put the reply to the reviewers. So please check and correct.
Extensive editing of the English language required
Author Response
Dear Editor and Reviewers,
In the attached file you can find the cover letter together with the point-to-point responses to the three reviewers' requests.
The text has been revised by a native English speaker to correct mistakes and bad constructed sentences.
Kind regards,
Luca Giorgio Bellucci

Round 4
Reviewer 3 Report
The authors provided the reply file to comments.